# Silencing the *Tlr4* Gene Alleviates Methamphetamine-Induced Hepatotoxicity by Inhibiting Lipopolysaccharide-Mediated Inflammation in Mice

**DOI:** 10.3390/ijms23126810

**Published:** 2022-06-18

**Authors:** Li-Bin Wang, Li-Jian Chen, Qi Wang, Xiao-Li Xie

**Affiliations:** 1Department of Toxicology, School of Public Health (Guangdong Provincial Key Laboratory of Tropical Disease Research), Southern Medical University, No. 1838 North Guangzhou Road, Guangzhou 510515, China; wanglibin20020202@163.com; 2Department of Forensic Pathology, School of Forensic Medicine, Southern Medical University, No. 1838 North Guangzhou Road, Guangzhou 510515, China; jianr2r@163.com

**Keywords:** methamphetamine, hepatotoxicity, inflammation, TLR4 pathway

## Abstract

Methamphetamine (METH) is a stimulant drug. METH abuse induces hepatotoxicity, although the mechanisms are not well understood. METH-induced hepatotoxicity was regulated by TLR4-mediated inflammation in BALB/c mice in our previous study. To further investigate the underlying mechanisms, the wild-type (C57BL/6) and *Tlr4^−/−^* mice were treated with METH. Transcriptomics of the mouse liver was performed via RNA-sequencing. Histopathological changes, serum levels of metabolic enzymes and lipopolysaccharide (LPS), and expression of TLR4-mediated proinflammatory cytokines were assessed. Compared to the control, METH treatment induced obvious histopathological changes and significantly increased the levels of metabolic enzymes in wild-type mice. Furthermore, inflammatory pathways were enriched in the liver of METH-treated mice, as demonstrated by expression analysis of RNA-sequencing data. Consistently, the expression of TLR4 pathway members was significantly increased by METH treatment. In addition, increased serum LPS levels in METH-treated mice indicated overproduction of LPS and gut microbiota dysbiosis. However, antibiotic pretreatment or silencing *Tlr4* significantly decreased METH-induced hepatic injury, serum LPS levels, and inflammation. In addition, the dampening effects of silencing *Tlr4* on inflammatory pathways were verified by the enrichment analysis of RNA-sequencing data in METH-treated *Tlr4^−/−^* mice compared to METH-treated wild-type mice. Taken together, these findings implied that *Tlr4* silencing, comparable to antibiotic pretreatment, effectively alleviated METH-induced hepatotoxicity by inhibiting LPS-TLR4-mediated inflammation in the liver.

## 1. Introduction

Methamphetamine (METH) is a stimulant that acts on the central nervous system, similar to amphetamine [1,2,3]. In addition to neurotoxicity, METH abuse results in hepatotoxicity and cardiovascular system abnormalities, among other organ injuries [4,5]. In clinical cases, METH intoxication causes hyperthermia and hyperammonemia, leading to acute hepatic failure [6]. Moreover, METH treatment impairs liver metabolism by disrupting the CYP1A2 metabolic pathway [7]. It also induces hepatotoxicity by arresting the cell cycle, inhibiting cell division, and activating apoptosis and autophagy [5,8]. The promotion of apoptosis restricts inflammation [9]. Long-term METH administration induces liver injury by activating oxidative stress and fibrosis [10]. In addition, METH-induced mitochondrial respiratory damage results in the accumulation of reactive oxygen species (ROS), which leads to oxidative stress and hepatotoxicity [11]. Additionally, METH treatment increases the expression of proinflammatory cytokines responsible for liver inflammation and neuroinflammation [12,13]. Nevertheless, the precise mechanisms underlying METH-induced liver damage are largely unknown.

Meanwhile, intestinal dysfunction is thought to participate in liver injury via the gut–liver axis [14]. The gut–liver axis is the bidirectional association between the gut (containing gut microbiota) and the liver, which is influenced by diet, drugs and environmental factors [15]. The gut microbial profile has been associated with inflammation in the liver, both in both clinical and animal studies [16,17]. Furthermore, metabolites of gut microbiota such as lipopolysaccharide (LPS) are related to liver inflammation [18,19]. LPS can be captured by pattern recognition receptors of the immune system, specifically by Toll-like-receptor 4 (TLR4), and then induce proinflammatory cascades mediated by cytokines [14]. In our previous study, TLR4-mediated inflammation was found to play an important role in METH-induced hepatotoxicity [20]. In addition, intestinal probiotics were reduced and opportunistic pathogens were increased by METH treatment in mice [21,22,23,24]. Antibiotics have been reported to alleviate liver injury induced by intestinal bacterial overgrowth [25]. In the present study, to further explore the role of antibiotics and the TLR4 pathway in METH-induced hepatotoxicity, antibiotic treatment was performed to evaluate the effect of METH administration in wild-type (C57BL/6J) and *Tlr4^−/−^* mice to provide basic experimental data for the treatment of METH-induced hepatotoxicity. In addition, RNA-sequencing was conducted to screen differentially expressed genes between the METH-treated wild-type and *Tlr4^−/−^* mice as well as their controls to further explore the underlying mechanisms.

## 2. Results

### 2.1. METH Induced Hepatotoxicity in Wild-Type Mice, Which Was Attenuated by Antibiotic Pretreatment

Histopathological changes, including shrunken nuclei, loose cytoplasm and extensive vacuolar degeneration, were observed in the livers of METH-treated mice (Figure 1A,B), and were improved by antibiotic pretreatment (Figure 1A,B). There were no histopathological changes in the antibiotic alone group (Figure 1A). Moreover, there were no significant differences in relative liver weights among the four groups (Figure 1C). Consistent with histopathological changes, AST and ALT serum levels were significantly elevated in METH-treated mice (Figure 1D), implying abnormal liver functions. However, AST and ALT levels were significantly suppressed by antibiotic pretreatment (Figure 1D). In addition, serum LPS levels significantly increased in METH-treated mice, while the level was significantly decreased by antibiotic pretreatment (Figure 1D). These results suggested that METH induced obvious hepatotoxicity, which was attenuated by antibiotic pretreatment.

### 2.2. Inflammatory Pathways Were Enriched in the Livers of METH-Treated Wild-Type Mice

Liver mRNA samples from the control and METH-treated groups were clustered using PCA analysis (Figure 2A). Using DESeq2, a total of 1772 differentially expressed genes (DEGs, 835 upregulated and 937 downregulated genes) were screened in METH-treated mice compared to the control (Figure 2B) to display the effects of METH treatment in wild-type mice. The top 10 up- and downregulated DEGs in the METH-treated group were listed in the Appendix A. The DEGs were annotated into different pathways by KEGG functional annotation analysis in METH-treated mice (Figure 2C). Through KEGG enrichment analysis, 27 significantly different pathways were enriched in METH-treated mice (Figure 2D). In these pathways, inflammatory mediator regulation of TRP channels, the Toll-like receptor signaling pathway, and the TNF signaling pathway were correlated with the inflammatory process, suggesting the involvement of inflammatory pathways in METH-induced hepatotoxicity. In addition, retinol metabolism, steroid hormone biosynthesis, fatty acid degradation, chemical carcinogenesis, drug metabolism-other enzymes, PPAR and AMPK signaling pathways were also enriched in METH-treated wild-type mice (Figure 2D).

### 2.3. METH Treatment Aroused Liver Inflammation by Activating the TLR4 Pathway, Which Was Suppressed by Antibiotic Pretreatment

Compared to the control group, the mRNA expression levels of *Tlr4*, *Myd88*, *Traf6*, *Rela*, *Il1b* and *Tnf* were significantly upregulated in the livers of METH-treated mice (Figure 3A). Moreover, the protein expression levels of TLR4, MyD88, TRAF6, NF-κB (p65), IL-1β and TNF-α were significantly elevated by METH treatment (Figure 3B), indicating the induction of inflammatory responses in the mouse liver. Antibiotic pretreatment significantly suppressed the expression levels of TLR4, MyD88, TRAF6, NF-κB (p65), TNF-α and IL-1β at both the mRNA (Figure 3A) and protein levels (Figure 3B).

### 2.4. Silencing of the Tlr4 Gene in Mice Significantly Ameliorated METH-Induced Liver Injury

Compared to METH-treated wild-type mice, METH-treated *Tlr4^−/−^* mice exhibited alleviated histopathological changes with slightly sparser cytoplasm and normal nuclei (Figure 4A,B). Furthermore, the relative liver weights were significantly decreased in the two *Tlr4^−/−^* groups compared to the control and METH-treated wild-type mice (Figure 4C), respectively. Serum AST, ALT, and LPS levels were significantly suppressed in METH-treated *Tlr4^−/−^* mice compared to the METH-treated wild-type mice (Figure 4D).

Moreover, the mRNA expression levels of *Myd88*, *Traf6*, *Rela*, *Il1b*, and *Tnf* as well as the protein expression levels of MyD88, TRAF6, NF-κB (p65), IL-1β, and TNF-α were significantly suppressed in METH-treated *Tlr4^−/−^* mice compared to wild-type mice (Figure 5), suggesting alleviated liver inflammation. In addition, no significant difference was determined among the groups in *Tlr4^−/−^* mice (Appendix A), although slight histopathological changes were observed in the livers of METH-treated *Tlr4^−/−^* mice (Figure 4 and Appendix A).

### 2.5. Enriched Inflammatory Pathways in METH-Treated Wild-Type Mice Were Regulated by Silencing Tlr4 in Mice

Samples from the METH-treated wild-type and *Tlr4^−/−^* groups were clustered by PCA analysis (Figure 6A). In the METH-treated *Tlr4^−/−^* group, 125 upregulated and 182 downregulated genes were screened out compared to the METH-treated wild-type group (Figure 6B) to demonstrate the effects of silencing the *Tlr4* gene. The top 10 up- and downregulated DEGs in the METH-treated *Tlr4^−/−^* group were listed in the Appendix A. The DEGs were annotated into different pathways via KEGG functional annotation analysis in the METH-treated *Tlr4^−/−^* group (Figure 6C). In addition, inflammatory mediator regulation of TRP channels (*p* < 0.001), Toll-like receptor signaling pathway (*p* < 0.001), NF-κB signaling pathway (*p* = 0.082) and TNF signaling pathway (*p* = 0.141) were identified in the METH-treated *Tlr4^−/−^* group through KEGG enrichment analysis (Figure 6D), suggesting the quenched inflammatory pathways by silencing *Tlr4* in attenuating METH-induced hepatotoxicity in *Tlr4^−/−^* mice. Interestingly, retinol metabolism, steroid hormone biosynthesis, fatty acid degradation, chemical carcinogenesis, drug metabolism, other enzymes, PPAR signaling pathway, and AMPK signaling pathway were also identified in METH-treated *Tlr4^−/−^* mice (Figure 6D), similar to METH-treated wild-type mice (Figure 2D). Moreover, 472 upregulated and 191 downregulated DEGs were screened out in the METH-treated *Tlr4^−/−^* group compared to the control (Appendix A), which was identified into 20 significantly different pathways (Appendix A) via KEGG enrichment analysis.

## 3. Discussion

METH has been demonstrated to induce hepatotoxicity by initiating oxidative stress, triggering inflammatory response, arresting cell cycle, and activating p53-mediated apoptosis and autophagy [5,8,20]. In the present study, we found that METH treatment induced obvious histopathological changes in the liver of wild-type mice and increased serum levels of AST and ALT, suggesting hepatic injury. Furthermore, inflammatory pathways associated with inflammatory mediator regulation of TRP channels, Toll-like receptor signaling pathway, NF-κB signaling pathway and TNF signaling pathway were enriched in METH-treated wild-type mice by KEGG enrichment analysis, suggesting the involvement of inflammatory pathways in METH-induced hepatic injury. Consistently, TLR4-mediated proinflammatory cytokines were significantly increased by METH treatment at both the mRNA and protein levels. However, pretreatment with antibiotics significantly attenuated METH-induced hepatic injury, including alleviating histopathological changes, decreasing the AST and ALT levels, and suppressing the expression of TLR4-mediated inflammatory cytokines, consistent with our previous study [20]. Different from the BALB/c mice in our previous study [20], C57BL/6 mice were used in the present study. Ascribing to the difference in mouse strain, METH-induced significant losses of the body and liver weights in BALB/c mice were not observed in C57BL/6 mice, although decreasing trends were observed. Nevertheless, METH-evoked inflammatory response was more robust in C57BL/6 mice.

In addition, alteration of gut microbiota, including increased conditioned pathogens and decreased probiotics, was observed in METH-treated mice in our previous study [23], resulting in the alteration of metabolites, such as overproduction of LPS. LPS is a prototypical pathogen/microbe-associated molecular pattern (P/MAMP) and can translocate to systemic sites [26]. The liver is the first organ that encounters microbial products, toxins and microorganisms from the intestine [27]. In addition to the alteration of flora, the intestinal barrier was also weakened by METH treatment [28], facilitating the translocation of intestinal contents, including the overproduction of LPS. In the liver, LPS can be captured by Toll-like-receptors (TLRs) and activate proinflammatory cascades [29]. METH induces liver injury by stimulating the expression and secretion of a series of proinflammatory cytokines [10,13]. In the present study, the serum level of LPS was significantly increased in METH-treated wild-type mice compared to the control, consistent with the increase in TLR4-mediated proinflammatory cytokines. In addition, multiple broad-spectrum antibiotics have been reported to attenuate hepatic injury induced by intestinal bacteria overgrowth [25], strengthen intestinal barrier functions, suppress serum endotoxin levels, and moderate liver inflammation [30]. Consistently, the METH-induced increase in the serum level of LPS was significantly suppressed in antibiotic-pretreated wild-type mice in the present study, suggesting decreased production of LPS and the associated flora, which might contribute to attenuating hepatic injury.

To further confirm the effects of TLR4-mediated inflammation on METH-induced hepatic injury, *Tlr4^−/−^* mice were used in the present study. Similar to the antibiotic-pretreatment in wild-type mice, histopathological changes, serum levels of AST, ALT, and LPS, and expression of proinflammatory cytokines were significantly alleviated in METH-treated *Tlr4^−/−^* mice compared to wild-type mice, indicating the significant role of the TLR4-mediated inflammatory pathway in this case. Although slight histopathological changes were observed in the livers of METH-treated *Tlr4^−/−^* mice, no significant difference was determined among the other groups in *Tlr4^−/−^* mice (Appendix A). Furthermore, several inflammatory pathways were identified in METH-treated *Tlr4^−/−^* mice (Figure 6D), confirming the dampening effect of silencing the *Tlr4* gene on inflammatory pathways.

In summary, our findings suggest that METH induced overproduction and translocation of LPS, consistent with our recent study, implying METH caused the increased abundance of pathogenic gut microbiota [28]. In the liver, LPS induces inflammation by activating the TLR4 pathway, which is one of the most important factors responsible for METH-induced hepatotoxicity. Pretreatment with antibiotics and silencing of the *Tlr4* gene alleviated METH-induced hepatotoxicity by improving gut microbiota dysbiosis and inhibiting inflammatory responses, respectively. These findings suggested that LPS-TLR4-mediated inflammation exerted an important role in METH-induced hepatotoxicity, which also took part in its toxic effects on mouse intestine [28]. In addition to LPS, different metabolites might be relevant to METH-induced hepatotoxicity, necessitating evaluation of the roles and mechanisms of metabolites in inducing hepatotoxicity.

## 4. Materials and Methods

### 4.1. Experimental Main Drugs

Methamphetamine (purity > 99%) was obtained from the National Institute Pharmaceutical and Biological Products Control (Beijing, China). Antibiotics, including neomycin, metronidazole, ampicillin and vancomycin, were purchased from Sangon Biotech (Shanghai, China).

### 4.2. Animals and Treatments

Wild-type mice (C57BL/6J, 6 weeks old, male) were provided by the Animal Laboratory Center of Southern Medical University. C57BL/10ScNJ mice (*Tlr4^−/−^*, TLR4-deficient mice) were purchased from GemPharmatech Co., Ltd. (Nanjing, China). All mice were housed in a standard SPF animal room at 24 ± 2 °C under 12 h light/dark cycle. The mice were given food and water ad libitum and acclimatized for 7 days before the formal experiment. The protocols for the animal experiments were approved by the ethics committee of Southern Medical University (Approval Code: L2018123) and performed in accordance with the guidelines of the National Institute of Health Guide for the Care and Use of Laboratory Animals of the same university.

The mice of each murine strain were randomly divided into four groups (10 mice per group): the control, METH, Abx + METH and Abx groups (Figure 7). The mice were given 6 intraperitoneal (i.p.) METH injections were administered at doses of 1.5, 4.5 and 7.5 mg/kg once a day for two days per dose, and then 8 injections were administered at a dose of 10.0 mg/kg four times per day at 2 h intervals [31]. Some of these mice were treated with a mix of antibiotics (Abx), including neomycin (200 mg/kg), metronidazole (200 mg/kg), ampicillin (200 mg/kg), and vancomycin (100 mg/kg) by gavage once a day. Abx was pretreated for 15 days [32,33,34], while METH was started on the 8th day. Mice were deep anesthetized and sacrificed 24 h after the last injection. Blood was quickly collected for serological analyses. Part of the liver tissues from 3 hepatic lobules was fixed in 10% formalin solution, while the other part was quickly frozen in liquid nitrogen and stored at −80 °C for further analysis.

### 4.3. Measurement of Serological Indicators

To collect serum, the blood was centrifuged for 15 min (3000 *g*) at 4 °C. The concentrations of aspartate aminotransferase (AST), alanine aminotransferase (ALT), proinflammatory cytokines (TNF-α, IL-1β, and IL-18) and LPS in the serum were determined using ELISA kits (Jiangsu Meibiao Biological Technology, Yancheng, China) according to the manufacturer’s instructions.

### 4.4. Observation of Histopathological Changes

After fixation, liver tissues were embedded in paraffin, sectioned into 3 μm pieces and stained using hematoxylin and eosin (H&E). Histopathological changes were scored by 3 independent pathologists who were not included in our experimental team according to the International Harmonization of Nomenclature and Diagnostic Criteria for Lesions in Rats and Mice [35].

### 4.5. Quantitative RT-PCR (RT-qPCR)

Total RNA was extracted from the liver tissues using a TRIzol reagent kit (Gaithersburg, MD, USA) according to the manufacturer’s instructions. Then, the RNA (2 μg per sample) was reverse transcribed into cDNA using HifairTM II first-strand cDNA synthesis SuperMix (Shanghai, China). Quantitative PCR was conducted using HieffTM qPCR SYBRR Green Master Mix (Shanghai, China) with a LightCycler^®^ 96 System (Roche Life Science, Penzberg, Germany). Primers used in the amplification were designed using PrimerBank (https://pga.mgh.harvard.edu/primerbank/, accessed on 1 March 2020). The primer sequences are exhibited in Table 1. β-Actin mRNA was used as the internal control.

### 4.6. Western Blotting Analysis

Liver tissues were ground into a uniform paste, ruptured using an ultrasonic disintegrator, and lysed using RIPA buffer supplemented with 1% protease inhibitor. The concentration of the total protein was determined using the Pierce™ BCA Protein Assay Kit (Waltham, MA, USA). Then, the total proteins (20 µg per sample) were separated using sodium dodecyl sulfate (SDS)-polyacrylamide gel electrophoresis (PAGE), transferred onto polyvinylidine difluoride membranes (BIO-RAD, California, USA), blocked using 5% skim milk for 2 h, and incubated with primary antibodies overnight at 4 °C. The membranes were incubated with secondary antibodies for 2 h at room temperature. Primary antibodies against TLR4 (Sc-293072, diluted 1:1000), MyD88 (Sc-74532, diluted 1:1000), Traf6 (Sc-8409, diluted 1:1000), NF-κB p65 (Sc-8008, diluted 1:1000), IL1β (Sc-52012, diluted 1:1000) and TNF-α (Sc-52746, diluted 1:1000) were obtained from Santa Cruz Biotechnology, Inc. (Santa Cruz, CA, USA) and had been validated in previous studies [36,37,38,39]. Anti-β-actin (8H10D10) antibody was purchased from Cell Signaling Technology Inc. (Boston, MA, USA).

The bands were visualized by enhanced chemiluminescence (BIO-RAD) and quantified using ImageJ software (version 1.52a) based on the integrated density values (IDVs). β-Actin was used as the internal control. The relative IDVs were presented as bar charts. The control group was designated a relative IDV of 1.

### 4.7. RNA-Sequencing (RNA-Seq) and Bioinformatics Analysis

RNA-sequencing was performed by Shanghai Majorbio Co., Ltd. (Shanghai, China). Sample collection and preparation were conducted according to the manufacturer’s recommended protocol. Briefly, total RNA of liver tissues from the wild type and *Tlr4^−/−^* groups (the control and METH treatment) was extracted using TRIzol^®^ Reagent (Invitrogen), according to the manufacturer’s instructions. The RNA quality and quantity were determined using a 2100 Bioanalyzer (Agilent) and ND-2000 NanoDrop, respectively. The transcriptome library was constructed using a TruSeq TMRNA sample preparation kit (Illumina, San Diego, USA). Briefly, mRNA was isolated and transcribed into double-stranded cDNA using a cDNA synthesis kit (Invitrogen, Carlsbad, CA, USA). The cDNA was repaired, phosphorylated, attached to ‘A’ bases, enriched and screened using PCR and beads for construction of the DNA Library. After quantification, the DNA was sequenced using an Illumina HiSeq xten/NovaSeq 6000 sequencer. The raw paired end reads were filtered using SeqPrep (https://github.com/jstjohn/SeqPrep, accessed on 1 December 2020) and Sickle (https://github.com/najoshi/sickle, accessed on 1 December 2020). The clean data were mapped to the reference genome using HISAT2 (http://ccb.jhu.edu/software/hisat2/index.shtml, accessed on 1 December 2020) and StringTie (https://ccb.jhu.edu/software/stringtie/index.shtml?t=example, accessed on 1 December 2020). Gene expression was quantified using RSEM (http://deweylab.biostat.wisc.edu/rsem/, accessed on 1 December 2020). The principal component analysis (PCA) was conducted to verify whether the samples could be separated into different groups. Differentially expressed genes (DEGs) were screened based on clean read counts. Significant expression was based on DESeq2 Q value ≤ 0.05 and |log2FC| > 1. Using the KEGG database, DEGs were classified into functional pathways. In addition, KEGG pathway enrichment analysis was conducted to identify significantly different metabolic pathways. The DEGs were significantly enriched at a Bonferroni-corrected *p*-value ≤ 0.05 compared with the whole-transcriptome background by KOBAS (http://kobas.cbi.pku.edu.cn/home.do, accessed on 1 December 2020). To better understand the effects of METH treatment and the silencing *Tlr4*, the two wild-type groups, METH-treated wild type and *Tlr4^−/−^* mice, as well the *Tlr4^−/−^* two groups were analyzed and compared in pairs, respectively.

### 4.8. Statistical Analysis

The data were presented as the mean ± SD. Statistical analysis was performed using GraphPad Prism (version 7.0, GraphPad Software, San Diego, CA, USA). Comparisons between two groups were conducted using two-sided Student’s *t* tests. One-way ANOVA followed by Tukey’s post hoc test was used to calculate statistical significance in multiple comparisons. A chi-square test was used to compare categorical variables in histological changes. *p* values < 0.05 were considered statistically significant.

## 5. Conclusions

Silencing the *Tlr4* gene effectively alleviated METH-induced hepatotoxicity by inhibiting LPS-TLR4-mediated inflammation in mice liver in a way comparable to antibiotic pretreatment.

## Figures and Tables

**Figure 1 ijms-23-06810-f001:**
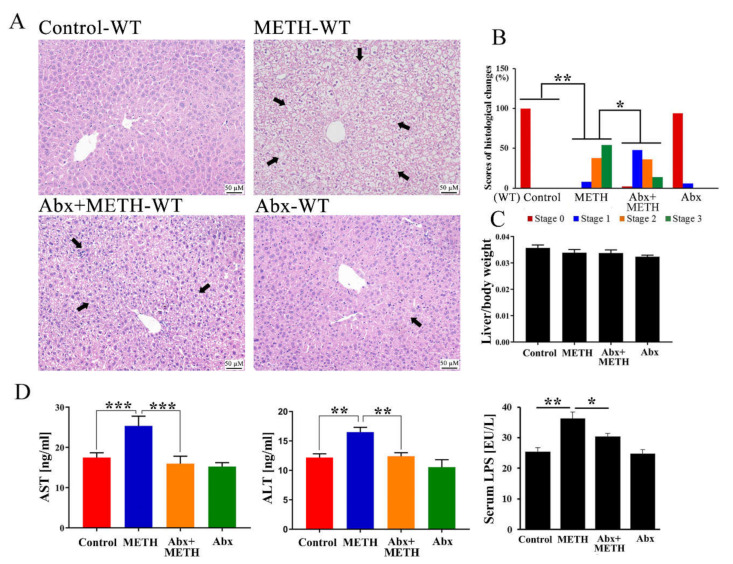
Histopathological changes, relative liver weight, and serum levels of AST, ALT and LPS. (**A**) H&E staining revealed shrinkage of the nuclei, loose cytoplasm and extensive vacuolar degeneration of hepatocytes (arrows) following METH treatment. However, pretreatment with antibiotics alleviated these histopathological changes. Bar = 50 µM. (**B**) The scores of histological changes. (**C**) The relative liver weight following METH treatment. There was no significant change among groups. (**D**) The effect of METH treatment on serum AST, ALT, and LPS levels. METH treatment increased the serum AST, ALT, and LPS levels. However, treatment with antibiotics reversed this phenomenon. Abx, antibiotics. * *p* < 0.05, ** *p* < 0.01, *** *p* < 0.001.

**Figure 2 ijms-23-06810-f002:**
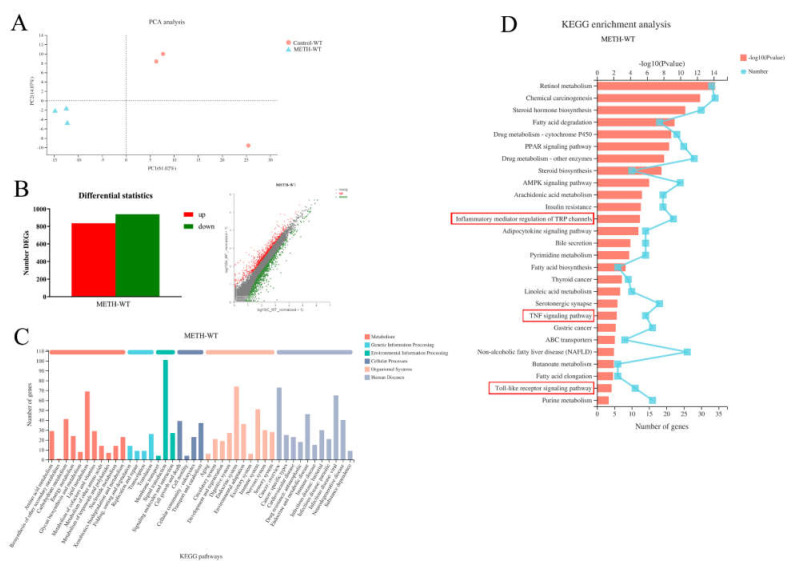
Inflammatory pathways were enriched in the mouse liver of the METH-treated group compared to the control by RNA-Seq analysis. (**A**) Principal component analysis; (**B**) Identification of DEGs using the histogram and scatter plot methods (|log2FC| ≥ 1, *p*-value < 0.05) in METH-treated wild-type (WT) group; (**C**) KEGG functional annotation analysis for metabolic pathways; (**D**) KEGG enrichment analysis in METH-treated WT mice. Inflammatory mediator regulation of TRP channels, the Toll-like receptor signaling pathway and the TNF signaling pathway (in the red boxes) were enriched and correlated with the inflammatory process.

**Figure 3 ijms-23-06810-f003:**
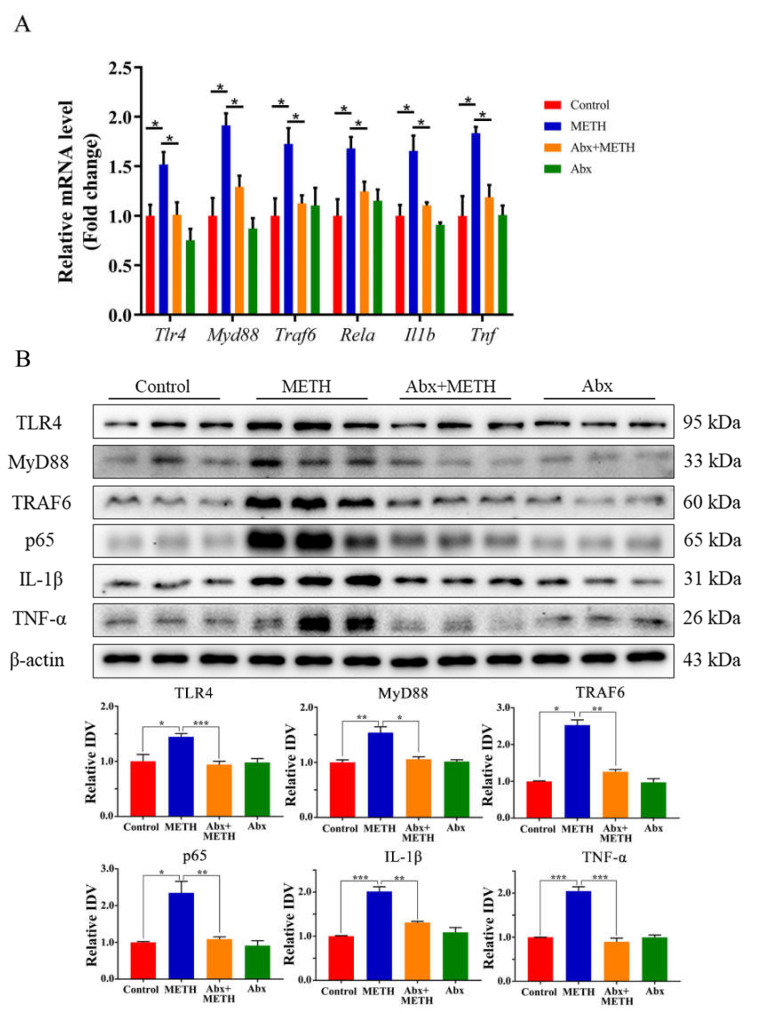
Expression of TLR4, MyD88, TRAF6, p65, IL-1β, and TNF-α at mRNA and protein levels in mouse liver. (**A**) Relative mRNA expression levels of *Tlr4*, *Myd88*, *Traf6*, *Rela*, *Il1b* and *Tnf* in the mouse liver. METH treatment upregulated the mRNA expression of *Tlr4*, *Myd88*, *Traf6*, *Rela*, *Il1b* and *Tnf*, but antibiotic pretreatment inhibited the effect of METH treatment on the aforementioned mRNAs. (**B**) The expression of TLR4, MyD88, TRAF6, p65, IL-1β and TNF-α proteins in mouse liver. Abx, antibiotics. * *p* < 0.05, ** *p* < 0.01, *** *p* < 0.001.

**Figure 4 ijms-23-06810-f004:**
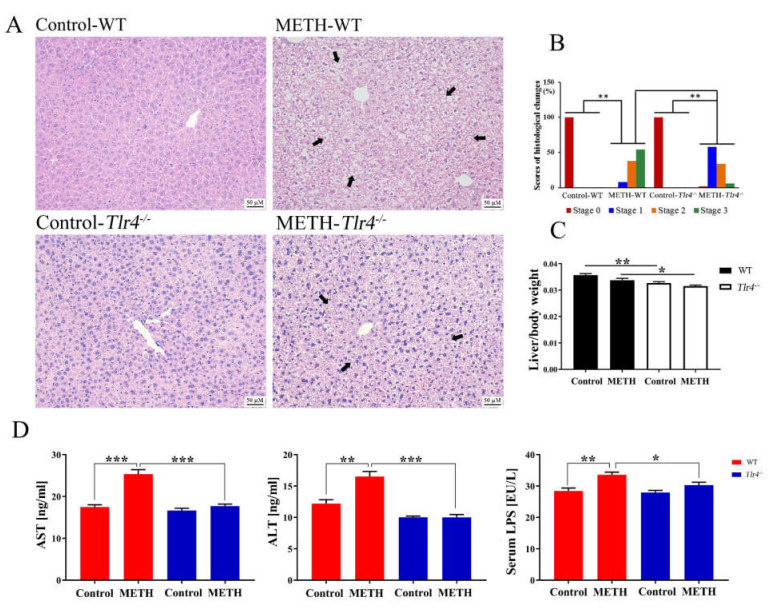
Silencing the *Tlr4* gene in mice significantly ameliorated METH-induced hepatic injury. (**A**) METH-induced histopathological changes (shrunken nucleus, loose cytoplasm, and extensive vacuolar degeneration; arrows) were remarkably weakened in *Tlr4^−/−^* mice. Bar = 50 µM. (**B**) The scores of histological changes; (**C**) The relative liver weights of the wild-type (WT) and *Tlr4^−/−^* mice; (**D**) Differences in serum AST, ALT, and LPS levels between the wild-type and *Tlr4^−/−^* mice. * *p* < 0.05, ** *p* < 0.01, *** *p* < 0.001.

**Figure 5 ijms-23-06810-f005:**
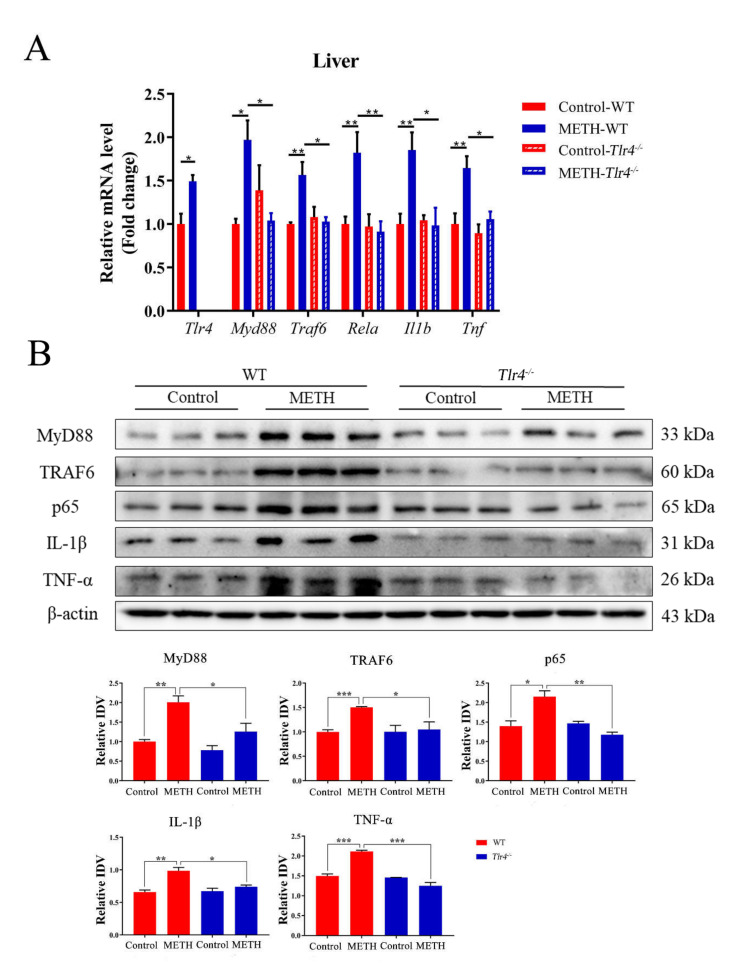
Expression of MyD88, TRAF6, p65, IL-1β, and TNF-α at mRNA and protein levels in mouse liver. (**A**) Relative mRNA expression of *Tlr4*, *Myd88*, *Traf6*, *Rela* (*p65*), *Il1b* and *Tnf* in wild-type and *Tlr4^−/−^* mice; (**B**) Protein expression of MyD88, TRAF6, p65, IL-1β and TNF-α in mouse liver. * *p* < 0.05, ** *p* < 0.01, *** *p* < 0.001.

**Figure 6 ijms-23-06810-f006:**
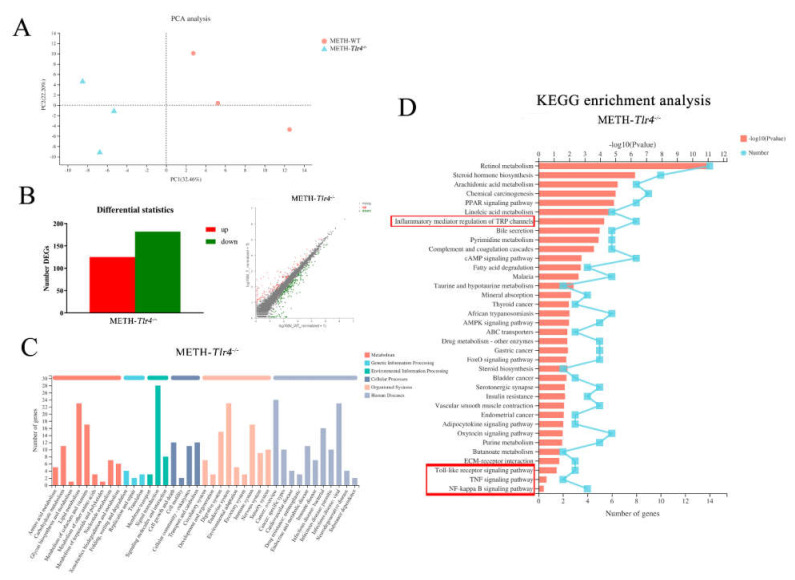
Inflammatory pathways were identified in the METH-treated *Tlr4^−/−^* group by RNA-Seq analysis. (**A**) Principal component analysis; (**B**) Identification of DEGs using the histogram and scatter plot methods (|log2FC| ≥ 1, *p*-value < 0.05) in the METH-treated *Tlr4^−/−^* group compared to the METH-treated WT group. (**C**) KEGG functional annotation analysis for metabolic pathways. (**D**) KEGG enrichment analysis in the METH-treated *Tlr4^−/−^* mice. Inflammatory mediator regulation of TRP channels, Toll-like receptor signaling pathway, NF-κB signaling pathway and TNF signaling pathway were identified in METH-treated *Tlr4^−/−^* group.

**Figure 7 ijms-23-06810-f007:**
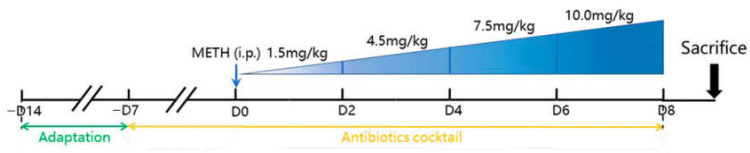
Experimental design.

**Table 1 ijms-23-06810-t001:** The primers for RT-qPCR.

Gene Name	Forward (5′-3′)	Reverse (5′-3′)
*Tlr4*	ATGGCATGGCTTACACCACC	GAGGCCAATTTTGTCTCCACA
*Myd88*	TCATGTTCTCCATACCCTTGGT	AAACTGCGAGTGGGGTCAG
*Traf6*	ATGCAGAGGAATCACTTGGCA	ACGGACGCAAAGCAAGGTT
*Rela (p65)*	AGGCTTCTGGGCCTTATGTG	TGCTTCTCTCGCCAGGAATAC
*Tnf*	CTGAACTTCGGGGTGATCGG	GGCTTGTCACTCGAATTTTGAGA
*Il1b*	GAAATGCCACCTTTTGACAGTG	TGGATGCTCTCATCAGGACAG
*Actb*	GGCTGTATTCCCCTCCATCG	CCAGTTGGTAACAATGCCATGT

## Data Availability

Data is contained within the article and its Appendix A.

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
