# Peer review of "Silencing the Tlr4 Gene Alleviates Methamphetamine-Induced Hepatotoxicity by Inhibiting Lipopolysaccharide-Mediated Inflammation in Mice"

_ijms, 2022, doi:10.3390/ijms23126810_

Round 1

Reviewer 1 Report

In this investigation, the authors have investigated the effects of methamphetamine on liver homeostasis and inflammation in two separate models. They first demonstrated that antibiotics alleviated meth-induced liver damage and inflammation. They went on to show that meth-induced damage is mediated through TLR4 by utilizing a knockout mouse model. The data  in Figs 1, 3, and 4 are clear, internally consistent, and support their overall conclusions. That the representative immunoblots in Figs 3 and 4 are all from the same blots is impressive, given the number of comparisons. However, there are several points that the authors should address to improve the manuscript.

Major points

  1. More details on the METH and antibiotics doses should be provided. The authors state that 8 doses were given, which to me implies a total number of doses; however, the reference cited (Bai et al., ref 29) specifies multiple injections (sometimes 4) per day. Please specify the exact dosing regimen. How often were antibiotics administered per day?
  1. Regarding the RNA seq figures (2 and 5), more explanation (and perhaps labeling) would be helpful to assist the reader with interpretation of the data and to support the authors’ conclusions. First, the methodology for the principal component analysis is not described; it would be helpful to include details in the Methods section. Second, the authors have concluded that inflammatory pathways were enriched in the METH-treated groups, but this does not seem to be displayed clearly in the figures. Panel B in each figure shows the number of up- and down-regulated genes, but the x-axis is labelled “Control-Wt vs METH-WT”. Is this a normalization or subtraction? If so, it needs to be specified how those groups are being compared since only two bars are shown (4 would be necessary to show up- and down- regulated genes in each group). Panel C shows the number of genes in different categories, but again, any differences between control and METH groups are not clear. Panel D similarly does not show a comparison between groups, but rather the number of genes and their p-values. How these groups are being compared should be specified, because these figures are difficult to interpret. Line 90 seems to indicate that it was a normalization: “METH-treated compared to control,” but the display of the results in the figure should allow the reader to interpret the authors’ findings with less difficulty. Relabeling the figures, or including additional details in the legend and Results section should fix this issue. Line 88 should indicate that control mice were compared to METH treated mice if this was the case – it only specifies METH-treated mice. Similarly, I’m not sure that Figure 5 is consistent with the authors’ conclusions in lines 148-151 and 200-201. Figure 2 concluded that inflammatory transcripts were “enriched” (I assume upregulated compared to controls) in METH-treated animals. This makes sense and is consistent with Figure 3. However, it does not make sense that inflammatory pathways would be enriched in the METH-treated Tlr4-/- mice compared to the METH-treated wild type mice since inflammatory markers were suppressed in the former (Figure 4). Like line 88, line 143 only specifies one group – the METH-treated Tlr4 -/- group. Panels B, D, and C would make more sense if it were indeed only one group that was examined. In both of these figures, only 2 groups (instead of all 4) were compared- the first examines the effects of METH in wild type mice, the second examines the effect of Tlr4-/- in METH-treated animals. A brief note explaining the rationale for just these comparisons would be helpful. I see that Supp Fig 2 compares control, and METH-treated Tlr4-/- mice – this should be mentioned in the Results section. Overall, the qPCR, serum, histology, and protein expression data (Figs 1, 3, and 4) are far more compelling (and understandable) than the RNA seq analysis, which, as currently presented, does not add much to the manuscript.
  1. I would recommend splitting Figure 4 into 2 figures, and present the ABX and Tlr4-/- results in the exact same format. It is very difficult to read the axes of the protein quantification graphs in Fig 4, even when magnifying to 200%. Furthermore, the histology in Fig 4 is difficult to see – it is much clearer in Figure 1. Panels E and F of Figure 4 should be separated to create a separate figure that matches Figure 3. The remaining data in Figure 4 should be rearranged to match Figure 1. Also, please define ABX in the text and figure legend. I assume it means “antibiotics.”
  1. The primary antibodies used for the immunoblot data should be validated, or the authors should provide references for their validation (not from the manufacturer) from previous studies. Primary antibody dilutions should be listed in the Methods. If not available in a reference, figures should be generated for each protein which show the entire lane (all molecular weights), demonstrating any non-specific banding or protein modifications. The authors should use proven positive and negative controls, so this would result in showing two lanes for each protein. These data could be provided in a supplemental file. Molecular weights of the proteins should be indicated in the figures.
  1. Line 207 – the authors suggest that gut microbiota dysbiosis was improved by antibiotics. I’m having trouble with this claim because dysbiosis was not assessed, and antibiotics are typically used to eliminate gut microbes, rather than improve them. If there is something about this antibiotic cocktail that targets pathogenic, rather than commensal bacteria, it should be pointed out. Otherwise, the sentence should be revised.
  1. It would be helpful to describe how the statistics were performed on the histological scores. Given the different scoring levels, I’m not sure that a one-way ANOVA is the correct test here.
  1. In the Introduction and Discussion, it would be helpful if the effects of METH on intestinal barrier function (in addition to flora) were specified (if known).
  1. In either the Introduction or Discussion, it should be made clear how this study is different from the authors’ previous study (Chen et al. Frontiers). I see that C57’s were used in the present study and Balbs in the previous. It appears that more robust inflammatory effects of METH were observed in the C57’s (especially Traf 6) compared to the Balbs in the previous study (Fig 8). If this is a species, rather than a dosing difference, it deserves some mention.

Minor

  1. When referring to the knockout mouse, lower-case, italic letters should be used (i.e. Tlr4-/-, not TLR4-/-). Upper case letters refer to the protein. These changes should be made throughout the manuscript.
  2. It should be specified somewhere that Rel A is p65.
  3. Line 20 – please indicate what condition it was that enriched the pathways.
  4. Lines 25-26 seem to indicate upregulation in METH-treated Tlr4 -/- mice see point 2 above.
  5. Lines 37-38 – This sentence does not make sense. Cells in the quiescent liver are not proliferating, so how would arresting the cell cycle induce hepatoxicity? Inhibiting cell division is the same thing as arresting the cell cycle. The authors should also specify that *excessive* apoptosis and autophagy could lead to hepatotoxicity since these processes could also be helpful to limit processes such as inflammation.
  6. Lines 50-52 – a phrase should be added that LPS signals specifically through TLR4.
  7. Lines 57-61 – this sentence is a run-on and does not make sense. Why are antibiotics and RNA seq paired together here?
  8. Line 68 and 134: change “shrank” to “shrunken”
  9. Line 72- should indicate that AST and ALT were suppressed in the METH treated ABX group.
  10. Line 75 – I would be more definitive here and change “might be” to “was.”
  11. Line 93 – I don’t understand the statement that the transcripts were correlated with the inflammatory process. Unless the authors mean the data presented in Fig 3, it does not appear that such a correlation was performed.
  12. Line 158 – include “METH treated” to describe the wild-type group.
  13. Line 288- I think “constriction” should be “construction”
  14. Figure 1 – font size in panels B, C, and D should be increased. Please define ABX in the figure legend.
  15. Figure 3 – font size should be increased in the protein expression graphs. Please define ABX in the figure legend.

Author Response

Response to Reviewer 1 Comments

In this investigation, the authors have investigated the effects of methamphetamine on liver homeostasis and inflammation in two separate models. They first demonstrated that antibiotics alleviated meth-induced liver damage and inflammation. They went on to show that meth-induced damage is mediated through TLR4 by utilizing a knockout mouse model. The data  in Figs 1, 3, and 4 are clear, internally consistent, and support their overall conclusions. That the representative immunoblots in Figs 3 and 4 are all from the same blots is impressive, given the number of comparisons. However, there are several points that the authors should address to improve the manuscript.

Major points

1. More details on the METH and antibiotics doses should be provided. The authors state that 8 doses were given, which to me implies a total number of doses; however, the reference cited (Bai et al., ref 29) specifies multiple injections (sometimes 4) per day. Please specify the exact dosing regimen. How often were antibiotics administered per day?

Response 1: We apologize for our carelessness. The total number of doses is 14. The exact dosing regimen has been added to the Materials and Methods Section, shown as “The mice were given 6 intraperitoneal (i.p.) METH injections were administered at doses of 1.5, 4.5, and 7.5 mg/kg once a day for two days per dose, and then 8 injections were administered at a dose of 10.0 mg/kg four times per day at 2 h intervals [31]. Some of these mice were treated with a mix of antibiotics (Abx), including neomycin (200 mg/kg), metronidazole (200 mg/kg), ampicillin (200 mg/kg), and vancomycin (100 mg/kg) by gavage once a day.”

2-1. Regarding the RNA seq figures (2 and 5), more explanation (and perhaps labeling) would be helpful to assist the reader with interpretation of the data and to support the authors’ conclusions. First, the methodology for the principal component analysis is not described; it would be helpful to include details in the Methods section.

Response 2-1: Thank you for your critical comments on this point. The principal component analysis (PCA), DEGs screening, KEGG annotation, and KEGG enrichment analyses were conducted on the software platform provided by the company of Shanghai Majorbio Co., Ltd (China). Accordingly, the following discussion was added in section 4.7. RNA-Sequencing (RNA-Seq) and bioinformatics analysis: “RNA-sequencing was performed by Shanghai Majorbio Co., Ltd (China). Sample collection and preparation were conducted according to the manufacturer’s recommended protocol. The principal component analysis (PCA) was conducted to verify whether the samples could be separated into different groups.”

2-2 Second, the authors have concluded that inflammatory pathways were enriched in the METH-treated groups, but this does not seem to be displayed clearly in the figures.

Response 2-2: Thank you for your critical comments on this point. The inflammatory pathways have been shown in the red boxes in Fig. 2D.

2-3 Panel B in each figure shows the number of up- and down-regulated genes, but the x-axis is labelled “Control-Wt vs METH-WT”. Is this a normalization or subtraction? If so, it needs to be specified how those groups are being compared since only two bars are shown (4 would be necessary to show up- and down- regulated genes in each group).

Response 2-3: Compared to the control group, the up- and down-regulated genes were shown in Fig. 2B. According to the suggestion, the label of the x-axis has been modified, shown as “METH-WT”.

2-4 Panel C shows the number of genes in different categories, but again, any differences between control and METH groups are not clear.

Response 2-4: The DEGs in METH group were annotated into pathways by KEGG annotation analysis (Fig. 2C). The number of genes in each pathway were shown in the y-axis, while the names of the detail pathways were presented in the x-axis.

2-5 Panel D similarly does not show a comparison between groups, but rather the number of genes and their p-values. How these groups are being compared should be specified, because these figures are difficult to interpret.

Response 2-5: By KEGG enrichment analysis, the DEGs in METH group were enriched in different pathways, as shown in Fig. 2D. The names of the detail pathways were displayed in the y-axis, while p value (-log10) and the number of genes were presented in the upper and lower x-axis, respectively. According to the suggestion, the legends have been modified.

2-6 Line 90 seems to indicate that it was a normalization: “METH-treated compared to control,” but the display of the results in the figure should allow the reader to interpret the authors’ findings with less difficulty. Relabeling the figures, or including additional details in the legend and Results section should fix this issue.

Response 2-6: According to the suggestion, the legends have been modified, shown as “Figure 2. Inflammatory pathways were enriched in the mouse liver of the METH-treated group compared to the control by RNA-Seq analysis. (A) Principal component analysis. (B) Identification of DEGs using the histogram and scatter plot methods (|log2FC| ≥ 1, p-value < 0.05) in METH-treated group. (C) KEGG functional annotation analysis for metabolic pathways. (D) KEGG enrichment analysis in METH-treated wild-type mice compared to the control. Inflammatory mediator regulation of TRP channels, the Toll-like receptor signaling path-way, and the TNF signaling pathway (in the red boxes) were enriched and correlated with the inflammatory process.”

2-7 Line 88 should indicate that control mice were compared to METH treated mice if this was the case – it only specifies METH-treated mice.

Response 2-7: According to the suggestion, the description of Fig.2 in the Results section has been modified, as follows: “Using DESeq2, a total of 1772 DEGs (835 upregulated and 937 downregulated genes) were screened in METH-treated mice compared to the control (Fig. 2B). The DEGs were annotated into different pathways by KEGG functional annotation analysis in METH-treated mice compared to the control (Fig. 2C).”

2-8 Similarly, I’m not sure that Figure 5 is consistent with the authors’ conclusions in lines 148-151 and 200-201. Figure 2 concluded that inflammatory transcripts were “enriched” (I assume upregulated compared to controls) in METH-treated animals. This makes sense and is consistent with Figure 3. However, it does not make sense that inflammatory pathways would be enriched in the METH-treated Tlr4-/- mice compared to the METH-treated wild type mice since inflammatory markers were suppressed in the former (Figure 4). Like line 88, line 143 only specifies one group – the METH-treated Tlr4 -/- group. Panels B, D, and C would make more sense if it were indeed only one group that was examined.

Response 2-8: By KEGG enrichment analysis, inflammatory pathways, including inflammatory mediator regulation of TRP channels, the Toll-like receptor signaling pathway, and the TNF signaling pathway, were enriched in METH-treated mice compared to the control (Fig. 2D). In original Fig. 5 (current Fig. 6), compared to the METH-treated wild-type mice, inflammatory mediator regulation of TRP channels, Toll-like receptor signaling pathway, NF-κB signaling pathway and TNF signaling pathway were enriched in METH-treated Tlr4-/- group, suggesting the regulation of inflammatory pathways by silencing Tlr4. When Tlr4 was silenced, the inflammatory markers were suppressed. Therefore, METH treatment could not significantly evoke inflammatory markers in Tlr4-/- mice compared to the Tlr4-/- control (Supplementary Fig.2). Moreover, compared to the METH-treated wild type mice, the inflammatory markers were downregulated (current Fig. 5). Consistently, based on the DEGs in METH-treated Tlr4-/- mice compared to METH-treated wild type mice, inflammatory pathways were enriched by KEGG enrichment analysis (current Fig. 6D).

2-9 In both of these figures, only 2 groups (instead of all 4) were compared- the first examines the effects of METH in wild type mice, the second examines the effect of Tlr4-/- in METH-treated animals.

Response 2-9: RNA-Seq analysis was conducted in four groups. The data of the four groups had been set in one figure for comparison. However, we found that it was confusing and inconvenient to interpret the main findings of this study. Therefore, the results of RNA-Seq analysis were divided into 3 parts, including METH-treated wild type mice compared to the control (Fig. 2), METH-treated Tlr4-/- mice compared to METH-treated wild type mice (current Fig. 6), and METH-treated Tlr4-/- mice compared to the Tlr4-/- control (Supplementary Fig.2).

2-10 A brief note explaining the rationale for just these comparisons would be helpful. I see that Supp Fig 2 compares control, and METH-treated Tlr4-/- mice – this should be mentioned in the Results section.

Response 2-10: According to the suggestion, a brief note explaining the rationale for just these comparisons has been added in the Materials and Methods section, shown as “To better understand the effects of METH treatment and the silencing Tlr4, the wild type two groups, METH-treated wild type and Tlr4-/- mice, as well the Tlr4-/- two groups were analyzed and compared in pairs, respectively.”

In addition, the results of Supp Figs 1 and 2 had been presented in the Discussion section, shown as “Although slight histopathological changes were observed in the livers of METH-treated Tlr4-/- mice, no significant difference was determined among the groups in Tlr4-/- mice (Supplementary Figures).” According to the suggestion, the results of Supp Fig. 2 have been added in the Results section, shown as “In addition, no significant difference was determined among the groups in Tlr4-/- mice (Supplementary Fig. 1), although slight histopathological changes were observed in the livers of METH-treated Tlr4-/- mice (Fig. 4 and Supplementary Fig. 1)” and “Moreover, 472 up-regulated and 191 down-regulated DEGs were screened in METH-treated Tlr4-/- group compared to the control (Supplementary Fig. 2B), which were enriched into 20 significantly different pathways (Supplementary Fig. 2D) by KEGG enrichment analysis.”

2-11 Overall, the qPCR, serum, histology, and protein expression data (Figs 1, 3, and 4) are far more compelling (and understandable) than the RNA seq analysis, which, as currently presented, does not add much to the manuscript.

Response 2-11: Thank you for your critical comments on this point. The DEGs screened out by RNA-Seq analysis were enriched into different pathways by KEGG enrichment analysis. In these pathways of METH-treated wild type group compared to its control, Inflammatory mediator regulation of TRP channels, Toll-like receptor signaling pathway and TNF signaling pathway were correlated with the inflammatory process (Fig. 2D), coinciding with elevated expression levels of inflammatory factors at the mRNA and protein levels, as shown in Fig. 3. Furthermore, consistent with the results presented in Fig. 4, inflammatory mediator regulation of TRP channels, Toll-like receptor signaling pathway, NF-κB signaling pathway and TNF signaling pathway were enriched in METH-treated Tlr4-/- group compared to METH-treated wild type mice (current Fig. 6D), suggesting the involvement of inflammatory pathways in attenuating METH-induced hepatotoxicity in Tlr4-/- mice. Therefore, the findings by RNA-Seq analysis displayed in Figs 2 and 6 corroborated the results in Figs 1, 3, 4, and 5. We hope our decision is acceptable, and we are ready and quite willing to hear further advice and suggestions if any.

3. I would recommend splitting Figure 4 into 2 figures, and present the ABX and Tlr4-/- results in the exact same format. It is very difficult to read the axes of the protein quantification graphs in Fig 4, even when magnifying to 200%. Furthermore, the histology in Fig 4 is difficult to see – it is much clearer in Figure 1. Panels E and F of Figure 4 should be separated to create a separate figure that matches Figure 3. The remaining data in Figure 4 should be rearranged to match Figure 1. Also, please define ABX in the text and figure legend. I assume it means “antibiotics.”

Response 3: According to the suggestion, Figure 4 has been split into two figures, Fig. 4 and Fig. 5. Accordingly, the original Figs. 5 and 6 have become into Figs. 6 and 7, respectively. The histology in Fig. 4 has been replaced by higher quality pictures. The protein quantification graphs have been instead by high quality graphs. The ABX has been defined in the Figure legends, and is presented as “Abx, antibiotics”.

4. The primary antibodies used for the immunoblot data should be validated, or the authors should provide references for their validation (not from the manufacturer) from previous studies. Primary antibody dilutions should be listed in the Methods. If not available in a reference, figures should be generated for each protein which show the entire lane (all molecular weights), demonstrating any non-specific banding or protein modifications. The authors should use proven positive and negative controls, so this would result in showing two lanes for each protein. These data could be provided in a supplemental file. Molecular weights of the proteins should be indicated in the figures.

Response 4: The primary antibodies had been used in our previous studies and other references. According to the suggestions, the references about the validation of the primary antibodies have been added in the Materials and Methods section. Primary antibody dilutions have been listed in the Methods. The molecular weights of the proteins have been indicated in the figures.

5. Line 207 – the authors suggest that gut microbiota dysbiosis was improved by antibiotics. I’m having trouble with this claim because dysbiosis was not assessed, and antibiotics are typically used to eliminate gut microbes, rather than improve them. If there is something about this antibiotic cocktail that targets pathogenic, rather than commensal bacteria, it should be pointed out. Otherwise, the sentence should be revised.

Response 5: Indeed, the antibiotic cocktail was used to eliminate gut microbes. The METH treatment decreased the diversity of probiotics but increased the abundance of pathogenic gut microbiota in our recent study [28], contributing to the disruption of the intestinal barrier. LPS could be overproduced in the case of gut microbiota dysbiosis and is related to liver inflammation. In the present study, serum LPS levels were significantly increased in METH-treated wild type mice compared to the control, while antibiotics pretreatment obviously suppressed the LPS levels. Therefore, we speculated that gut microbiota dysbiosis was improved by antibiotics pretreatment. According to the suggestion, the sentence has been rewritten, shown as “the METH-induced increase in the serum level of LPS was significantly suppressed in antibiotic-pretreated wild-type mice in the present study, suggesting the decreased production of LPS and the associated flora, which might contribute to attenuating hepatic injury.”

6. It would be helpful to describe how the statistics were performed on the histological scores. Given the different scoring levels, I’m not sure that a one-way ANOVA is the correct test here.

Response 6: According to the suggestion, the statistical method performed on the histological scores has been added to the Materials and Methods section, shown as “Chi-square test was used to compare categorical variables in histological changes.”

7. In the Introduction and Discussion, it would be helpful if the effects of METH on intestinal barrier function (in addition to flora) were specified (if known).

Response 7: According to the suggestion, the effects of METH on the intestinal barrier and flora have been added to the Discussion section, shown as “In addition to the alteration of flora, intestinal barrier was also weakened by METH treatment [28], facilitating the translocation of intestinal contents, including the overproduction of LPS.”

8. In either the Introduction or Discussion, it should be made clear how this study is different from the authors’ previous study (Chen et al. Frontiers). I see that C57’s were used in the present study and Balbs in the previous. It appears that more robust inflammatory effects of METH were observed in the C57’s (especially Traf 6) compared to the Balbs in the previous study (Fig 8). If this is a species, rather than a dosing difference, it deserves some mention.

Response 8: According to the suggestion, the comparison has been added to the Discussion section, presented as “Different from the BALB/c mice in our previous study [20], C57BL/6 mice were used in the present study. Ascribing to the difference in mouse strain, METH-induced significant losses of the body and liver weights in BALB/c mice were not observed in C57BL/6 mice, although the decreasing trends were observed. Nevertheless, METH-evoked inflammatory response was more robust in C57BL/6 mice.”

Minor

1. When referring to the knockout mouse, lower-case, italic letters should be used (i.e. Tlr4-/-, not TLR4-/-). Upper case letters refer to the protein. These changes should be made throughout the manuscript.

Response 1: According to the suggestion, the TLR4-/- has been replaced by Tlr4-/- throughout the manuscript.

2. It should be specified somewhere that Rel A is p65.

 Response 2: According to the suggestion, p65 has been inserted into Table 1.

3. Line 20 – please indicate what condition it was that enriched the pathways.

Response 3: According to the suggestion, the condition has been added, shown as “inflammatory pathways were enriched in the liver of METH-treated mice as demonstrated by expression analysis of RNA-sequencing data.”

4. Lines 25-26 seem to indicate upregulation in METH-treated Tlr4 -/- mice see point 2 above.

Response 4: Because the Tlr4 gene was silenced, inflammatory-associated indicators were significantly downregulated. Compared to the METH-treated wild type mice, the DEGs were enriched into different pathways by KEGG enrichment analysis in METH-treated Tlr4-/- mice, including inflammatory pathways. Therefore, the inflammatory pathways were constituted by downregualted genes. The findings coincided with the data of the original Fig. 4 (current Fig. 5).

5. Lines 37-38 – This sentence does not make sense. Cells in the quiescent liver are not proliferating, so how would arresting the cell cycle induce hepatoxicity? Inhibiting cell division is the same thing as arresting the cell cycle. The authors should also specify that *excessive* apoptosis and autophagy could lead to hepatotoxicity since these processes could also be helpful to limit processes such as inflammation.

Response 5: METH-induced cell cycle arrest and inhibition of cell division had been observed in our previous in vitro study [5], contributing to its hepatotoxicity. In addition, according to the suggestion, the sentence “The promotion of apoptosis restricts inflammation [9].” has been inserted.

6. Lines 50-52 – a phrase should be added that LPS signals specifically through TLR4.

Response 6: According to the suggestion, the description has been added, shown as “LPS can be captured by pattern recognition receptors of the immune system, specifically by Toll-like-receptor 4 (TLR4), and then induce proinflammatory cascades mediated by cytokines”.

7. Lines 57-61 – this sentence is a run-on and does not make sense. Why are antibiotics and RNA seq paired together here?

Response 7: According to the suggestion, the sentence has been rewritten, presented as “In the present study, to further explore the role of antibiotics and the TLR4 pathway in METH-induced hepatotoxicity, antibiotic treatment was performed to evaluate the effect of METH administration in wild-type (C57BL/6J) and Tlr4-/- mice to provide basic experimental data for the treatment of METH-induced hepatotoxicity. In addition, RNA-sequencing was conducted to screen differentially expressed genes between the METH-treated wild-type and Tlr4-/- mice as well as their controls to further explore the underlying mechanisms.”

8. Line 68 and 134: change “shrank” to “shrunken”

Response 8: According to the suggestion, “shrank” has been replaced by “shrunken”.

9. Line 72- should indicate that AST and ALT were suppressed in the METH treated ABX group.

Response 9: The description had been included in the text, shown as “However, AST and ALT levels were significantly suppressed by antibiotic pretreatment (Fig. 1D).”

10. Line 75 – I would be more definitive here and change “might be” to “was.”

Response 10: According to the suggestion, “might be” has been changed to “was”.

11. Line 93 – I don’t understand the statement that the transcripts were correlated with the inflammatory process. Unless the authors mean the data presented in Fig 3, it does not appear that such a correlation was performed.

Response 11: The DEGs were screened out by RNA-Seq analysis in METH-treated wild type mice compared to the control. The DEGs were enriched into different pathways by KEGG enrichment analysis, including inflammatory mediator regulation of TRP channels, the Toll-like receptor signaling pathway, and the TNF signaling pathway. These pathways were correlated with the inflammatory process. The findings were consistent with the data presented in Fig. 3.

12. Line 158 – include “METH treated” to describe the wild-type group.

Response 12: According to the suggestion, “METH-treated” has been added, presented as “Serum AST, ALT, and LPS levels were significantly suppressed in METH-treated Tlr4-/- mice compared to the METH-treated wild-type mice (Fig. 4D).”

13. Line 288- I think “constriction” should be “construction”

Response 13: We apologize for our carelessness. The mistake has been corrected.

14. Figure 1 – font size in panels B, C, and D should be increased. Please define ABX in the figure legend.

Response 14: According to the suggestion, the font size in panels B, C, and D of Figure 1 have been increased. ABX has been defined, shown as “Abx, antibiotics.”

15. Figure 3 – font size should be increased in the protein expression graphs. Please define ABX in the figure legend.

Response 15: According to the suggestion, the font size in the protein expression graphs have been increased. ABX has been defined, shown as “Abx, antibiotics.”

We appreciate the time and effort the reviewer has put to give us such critical and constructive comments to improve the manuscript. We have tried our best to modify our manuscript according to the suggestions, and we hope our decision as well as explanations are acceptable, and we are ready and quite willing to hear further advice and suggestions if any.

Reviewer 2 Report

The paper describes that Tlr4 silencing, effectively alleviates METH-induced hepatotoxicity by inhibiting LPS-TLR4-mediated inflammation in the liver of wild-type (C57BL/6) and TLR4-/- mice. It is a well-designed study that actually describes the continuation of team’s work this this field.  In my opinion, this is a well written manuscript and the work can be considered for publication after the above minor changes:

  • Authors must provide an explanation that there were no significant differences in the relative liver weights among the Abx, METH and Abx-METH treated of C57BL/6 (Fig. 1C). In the previous report [19] a significant decrease of the body and liver weights of BABL/c mice was recorded after Abx, METH, moreover, antibiotic pre-treatment tended to alleviate METH‐induced body and liver weight loss.
  • There is no comparison of the results in the 'Discussion Section' with analogous studies that report alleviation of METH-induced hepatotoxicity. Authors must include a short overview of the literature.
  • H&E staining images must be provided in better resolution, the font size of the scale bar must be increased and more importantly is the insertion of arrows that point to the nucleus and cytoplasm. In addition, Figs 1A & 4A must be presented in a similar manner employing the same titles (control or control-WT, etc)
  • 2, Fig 4B-D and Fig.5 must increase the font size of the inset titles and be provided in better resolution
  • The description that ''the effect of Tlr4 silencing is similar to antibiotic pretreatment'' is not clearly supporting by the experimental results, since in some cases Abx pre-treatment is more effective. Authors better use the word ‘comparable’ or ‘in a comparable way’

Author Response

Response to Reviewer 2 Comments

The paper describes that Tlr4 silencing, effectively alleviates METH-induced hepatotoxicity by inhibiting LPS-TLR4-mediated inflammation in the liver of wild-type (C57BL/6) and TLR4-/- mice. It is a well-designed study that actually describes the continuation of team’s work this this field.  In my opinion, this is a well written manuscript and the work can be considered for publication after the above minor changes:

1. Authors must provide an explanation that there were no significant differences in the relative liver weights among the Abx, METH and Abx-METH treated of C57BL/6 (Fig. 1C). In the previous report [19] a significant decrease of the body and liver weights of BABL/c mice was recorded after Abx, METH, moreover, antibiotic pre-treatment tended to alleviate METH‐induced body and liver weight loss.

Response 1: In the present study, C57BL/6 mice were used to perform experiments. Fig. 1C presents the relative liver weight. Treatment of METH, Abx, or Abx+METH tended to decrease the body and liver weights, with no significant differences compared to the control. In the previous study [20], BALB/c mice were used to study the effects of METH. The body weight and absolute liver weights of mice were significantly decreased by antibiotic or METH treatment. Antibiotic pretreatment tends to alleviate METH-induced body and liver weight loss, but no significant difference was observed, consistent with our present study. Different strains of mice might have different sensitivities to METH-induced losses of body and liver weights.

2. There is no comparison of the results in the 'Discussion Section' with analogous studies that report alleviation of METH-induced hepatotoxicity. Authors must include a short overview of the literature.

Response 2: According to the suggestions, a short overview of the literature has been included in the Discussion section, shown as “METH has been demonstrated to induce hepatotoxicity by initiating oxidative stress, triggering inflammatory response, arresting cell cycle, and activating p53-mediated apoptosis and autophagy [5, 8, 20].”

3. H&E staining images must be provided in better resolution, the font size of the scale bar must be increased and more importantly is the insertion of arrows that point to the nucleus and cytoplasm. In addition, Figs 1A & 4A must be presented in a similar manner employing the same titles (control or control-WT, etc)

Response 3: High quality H&E staining images have been provided (600 dpi, height and wide). The font size of the scale bar has been increased. The bar has also been defined in the Legends.

According to the suggestions, the insertion of arrows has been added to the images. Figs 1A and 4A have presented in a similar manner employing the same titles.

4. Fig 4B-D and Fig.5 must increase the font size of the inset titles and be provided in better resolution

Response 4: According to the suggestions, the font size of the inset titles in the original Figs 4 and 5 (current Figs 4-6) has been enlarged. The resolution has been improved (600 dpi, height and wide).

5. The description that ''the effect of Tlr4 silencing is similar to antibiotic pretreatment'' is not clearly supporting by the experimental results, since in some cases Abx pre-treatment is more effective. Authors better use the word ‘comparable’ or ‘in a comparable way’

Response 5: According to the suggestions, the “similar” has been replaced by “in a way comparable to”, shown as “Silencing the Tlr4 gene effectively alleviated METH-induced hepatotoxicity by inhibiting LPS-TLR4-mediated inflammation in mice liver in a way comparable to antibiotic pretreatment.” In the Abstract, the “similar” has been replaced by “comparable”, presented as “Taken together, these findings implied that Tlr4 silencing, comparable to antibiotic pretreatment, effectively alleviated METH-induced hepatotoxicity by inhibiting LPS-TLR4-mediated inflammation in the liver.”

Reviewer 3 Report

Wang et al. aimed to explore the role of antibiotics and the TLR4 pathway in METH-induced hepatotoxicity. Overall, this study is well written, follows a scientifically sound experimental design, and shows convincing and novel results. Other minor improvements to the manuscript are provided in the comments below. I suggest approval after a minor revision.

Overall, the subject is very interesting: investigation of the underlying mechanisms of METH-induced hepatotoxicity has the potential of great value in clinical practice. Accurate determination of the role of antibiotics and the TLR4 pathway in METH-induced hepatotoxicity and the effects of antibiotic treatment on the gut-liver axis is important for therapeutic decision-making in human studies. This study presents solid evidence on murine models and is therefore a ground point for future translational studies in humans. I have several minor issues with the work and experimental data presented, but am otherwise enthusiastic about the presentation, flow, and coverage of the manuscript:

Minor issues

i) Page 1, line 20: “… inflammatory pathways were enriched in the liver by RNA-sequencing.“ should read  “… inflammatory pathways were enriched in the liver as demonstrated by expression analysis or RNA-sequencing data.”

ii) Page 1, line 41: Add explanation for ROS, i.e. "reactive oxygen species", upon its first use in the manuscript.

iii) Page 4, Figure 2A and page 7 Figure 5B: As one of the study aims was to provide basic experimental data, I suggest authors add a table listing the top 10 up-and down-regulated genes from RNA-seq analysis. In addition to tested and genes shown in the results, one would also expect upregulation of, for instance, Il-6 and downregulated expression of Arg-1, Il-10, and KLF4. It would be interesting to show dose-dependent down-regulation of Il-10. In addition, I am wondering why the authors did not use TAK-242, an TLR4 antagonist/inhibitor as a negative control?

iv) Page 4, Figure 2A: The experimental design in this study defines 80 subjects: 2 murine strains (WT, Tlr4-/-), 4 groups per strain (C, M, Abx+M, Abx), and 10 subjects per group. Could the authors explain why are only 3 subjects per group shown in Fig. 2A? The same hesitation applies to Fig. 5A on page 7.

v) Page 8, line 210-211: Authors should consider inserting their recent publication (Wang et al., 2022 doi:/10.1016/j.taap.2022.116011) and commenting on intersecting findings and highlights from that study in the concluding section of the manuscript.

vi) All Figures: Re-align the figure with the main body of the text.

Author Response

Response to Reviewer 3 Comments

Wang et al. aimed to explore the role of antibiotics and the TLR4 pathway in METH-induced hepatotoxicity. Overall, this study is well written, follows a scientifically sound experimental design, and shows convincing and novel results. Other minor improvements to the manuscript are provided in the comments below. I suggest approval after a minor revision.

Overall, the subject is very interesting: investigation of the underlying mechanisms of METH-induced hepatotoxicity has the potential of great value in clinical practice. Accurate determination of the role of antibiotics and the TLR4 pathway in METH-induced hepatotoxicity and the effects of antibiotic treatment on the gut-liver axis is important for therapeutic decision-making in human studies. This study presents solid evidence on murine models and is therefore a ground point for future translational studies in humans. I have several minor issues with the work and experimental data presented, but am otherwise enthusiastic about the presentation, flow, and coverage of the manuscript:

Minor issues

1. i) Page 1, line 20: “… inflammatory pathways were enriched in the liver by RNA-sequencing.“ should read “… inflammatory pathways were enriched in the liver as demonstrated by expression analysis or RNA-sequencing data.”

Response 1: According to the suggestion, the sentence has been modified, presented as “inflammatory pathways were enriched in the liver of METH-treated mice as demonstrated by expression analysis of RNA-sequencing data.”

2. ii) Page 1, line 41: Add explanation for ROS, i.e. "reactive oxygen species", upon its first use in the manuscript.

Response 2: According to the suggestion, the explanation has been added, shown as “reactive oxygen species (ROS)”.

3. iii) Page 4, Figure 2A and page 7 Figure 5B: As one of the study aims was to provide basic experimental data, I suggest authors add a table listing the top 10 up-and down-regulated genes from RNA-seq analysis. In addition to tested and genes shown in the results, one would also expect upregulation of, for instance, Il-6 and downregulated expression of Arg-1, Il-10, and KLF4. It would be interesting to show dose-dependent down-regulation of Il-10. In addition, I am wondering why the authors did not use TAK-242, an TLR4 antagonist/inhibitor as a negative control?

Response 3: According to the suggestion, 2 tables have been added as Supplementary materials. In the present study, expressions of the TLR4 pathway components were focused. TAK-242 is a selective TLR4 inhibitor and has to be administered in an invasive manner in vivo study. Therefore, the Tlr4-/- mice were chosen to explore the important role of TLR4-mediated inflammation in METH-induced hepatotoxicity. We are grateful for the suggestion. TAK-242 is considered to use in our further in vitro study.

4. iv) Page 4, Figure 2A: The experimental design in this study defines 80 subjects: 2 murine strains (WT, Tlr4-/-), 4 groups per strain (C, M, Abx+M, Abx), and 10 subjects per group. Could the authors explain why are only 3 subjects per group shown in Fig. 2A? The same hesitation applies to Fig. 5A on page 7.

Response 4: RNA-Sequencing was conducted in four groups (3 mice per group), including wide type control and METH groups, Tlr4-/- control and METH groups. Therefore, there are 3 subjects per group in Figs. 2A and 5A (current 6A). The effects of METH treatment in wild type mice were analyzed by comparing METH-treated mice and the control, while the effects of silencing Tlr4 gene on METH-induced hepatotoxicity were analyzed by comparing METH-treated Tlr4-/- mice and wild type mice. The comparison between the Tlr4-/- two groups further verified that METH-induced hepatotoxicity was significantly alleviated by silencing Tlr4 gene.

5. v) Page 8, line 210-211: Authors should consider inserting their recent publication (Wang et al., 2022 doi:/10.1016/j.taap.2022.116011) and commenting on intersecting findings and highlights from that study in the concluding section of the manuscript.

Response 5: According to the suggestion, the reference has been added. The findings and highlights from that study have been commented in the Discussion section, presented as “In summary, our findings suggest that METH induced overproduction and translo-cation of LPS, consistent with our recent study, attributing to METH-caused the increased abundance of pathogenic gut microbiota [28]. In the liver, LPS induces inflammation by activating the TLR4 pathway, which is one of the most im-portant factors responsible for METH-induced hepatotoxicity. Pretreatment with antibiotics and silencing of the Tlr4 gene alleviated METH-induced hepatotoxicity by improving gut microbiota dysbiosis and inhibiting inflammatory responses, respectively. These findings suggested that LPS-TLR4-mediated inflammation exerted an important role in METH-induced hepatotoxicity, which also took a part in its toxicity effects on mouse intestine [28]. In addition to LPS, different metabolites might be relevant to METH-induced hepatotoxicity, necessitating evaluation of the roles and mechanisms of metabolites in inducing hepatotoxicity.”

6. vi) All Figures: Re-align the figure with the main body of the text.

Response 6: According to the suggestion, the figures have been realigned with the main body of the text.

Round 2

Reviewer 1 Report

I thank the authors for their detailed response and editing the manuscript. Some figures look better and methodological details have been provided. However, some of the issues that I brought up in my first review were not adequately addressed by the authors.

Updated Figure 2 shows that PCA separates the groups, and now seems to indicate that the enrichment analysis was only performed on one group of animals (the METH-WT). If panels B, C, and D reflect the data in only one group of animals, then I understand what it is showing. Unfortunately, in the Results text, the authors are still making a comparison between the groups when referring to Figure 2. What needs to be clearer is how the data in figures 2 and 6 show the comparison between groups (if they even do). I will also note that new Fig 6 is labelled in the same way as the initial submission- the authors did not modify Fig 6 like they did Fig 2, so now this is even more confusing. Axis units of “Number of DEGs,” “Number of genes”, or “p-values” do not indicate any kind of comparison between groups – this is my main point. Please elaborate on this for readers who might not be familiar with this technique. This should only require a sentence or two about the technique, which could be included in the Results text and the figure legend. The authors have not sufficiently clarified this issue and there is a disconnect between what they are claiming in the Results text, and what can be interpreted from the data in Figures 2 and 6. 

 For comparison, the results in Supp Table 1 are presented in a clear way.

Gene name

Regulate

FC (M_Tlr4-/-/M_WT)

P adjust

1

Gm28047

up

553.3

3.57E-11

If FC is a fold value, then this tells me that Gm28047 is 553-fold greater in the Tlr4-/- compared to the wild type.

Related, I’m having trouble with the authors’ use of the word, “enriched.” The authors need to be clear in their results whether transcripts are up- or downregulated between conditions and what particular data indicate these assertions.

 Regarding the authors' response: “When Tlr4 was silenced, the inflammatory markers were suppressed-” where exactly is this shown in the RNA seq results? Figure 6 does not clearly indicate this assertion, nor does the description in the Results.

I’m also having trouble with this statement: “In addition, inflammatory mediator regulation of TRP channels (p < 0.001), Toll-like receptor signaling pathway (p < 0.001), NF-κB signaling pathway (p = 0.082) and TNFsignaling pathway (p = 0.141) were enriched in METH-treated Tlr4-/- group through KEGG enrichment analysis compared to wild-type group (Fig. 6D), suggesting the regulation of inflammatory pathways by silencing Tlr4 in attenuating METH-induced hepatotoxicity in Tlr4-/- mice.” This statement still seems to imply that inflammatory pathways are upregulated (“enriched”) in Tlr4-/- mice. The word “regulation” in the phrase “suggesting the regulation of inflammatory pathways’” is vague. Something can be up- or downregulated. Please revise for clarity. Furthermore, some of these p-values do not reach a value of 0.05.

Abstract: “Moreover, inflammatory pathways were also significantly enriched in METH-treated Tlr4-/- mice compared to METH treated wild type mice suggesting the regulatory effect of Tlr4 silencing on inflammatory pathways.” This statement is confusing, and suggests to me that inflammatory pathways were upregulated in the Tlr4 knockouts, compared to the METH-treated wild type which seems contradictory. The authors have not adequately addressed this concern in either their response, or the revised manuscript. They need to elaborate on whether this is indeed the case. If it is, it would seem like a counterintuitive result that needs further explanation in the discussion section.

 Page 14: This revision does not make sense, please revise: “consistent with our recent study, attributing to METH-caused the increased abundance of pathogenic gut microbiota”

Overall, there are still points that need to be clarified so that the reader can properly interpret the authors’ data and so that it is clear that their data support their conclusions. These are changes that were requested in the first review and have not been sufficiently addressed.

Author Response

1. Updated Figure 2 shows that PCA separates the groups, and now seems to indicate that the enrichment analysis was only performed on one group of animals (the METH-WT). If panels B, C, and D reflect the data in only one group of animals, then I understand what it is showing. Unfortunately, in the Results text, the authors are still making a comparison between the groups when referring to Figure 2. What needs to be clearer is how the data in figures 2 and 6 show the comparison between groups (if they even do). I will also note that new Fig 6 is labelled in the same way as the initial submission- the authors did not modify Fig 6 like they did Fig 2, so now this is even more confusing. Axis units of “Number of DEGs,” “Number of genes”, or “p-values” do not indicate any kind of comparison between groups – this is my main point. Please elaborate on this for readers who might not be familiar with this technique. This should only require a sentence or two about the technique, which could be included in the Results text and the figure legend. The authors have not sufficiently clarified this issue and there is a disconnect between what they are claiming in the Results text, and what can be interpreted from the data in Figures 2 and 6.

 For comparison, the results in Supp Table 1 are presented in a clear way.

If FC is a fold value, then this tells me that Gm28047 is 553-fold greater in the Tlr4-/- compared to the wild type.

Response 1: Thank you for your constructive comment. As shown in Supplementary Table 1, the differentially expressed genes (DEGs) were screened out by comparing the two groups. In Fig. 2, the DEGs in the METH-treated WT group were obtained by comparing to the control group, displaying the effects of METH treatment. The axis units of “Number of DEGs” and “Number of genes” indicate the genes in the METH-treated WT group, while “p-value” indicates the significant enrichment of the pathway. Panels B, C, and D reflect the data in only one group of the METH-treated WT group. The descriptions of panels C and D in the Results section have been modified to reduce confusion. The subtitles of Fig. 2 had been modified in R1 revision.

In Fig. 6, the DEGs in the METH-treated Tlr4-/- group were screened out by comparing with the METH-treated WT group but not the Tlr4-/- control group, showing the effects of silencing the Tlr4 gene. Supplementary Fig. 2 shows the DEGs in the METH-treated Tlr4-/- group compared to the Tlr4-/- control group. Accordingly, the subtitles in Fig. 6 have been modified in this revision. The descriptions of panels C and D in the Results section have been modified to reduce confusion.

According to the suggestions, descriptions of the DEGs have been revised in the Results text, showing as “a total of 1772 differentially expressed genes (DEGs, 835 upregulated and 937 downregulated genes) were screened in METH-treated mice compared to the control (Fig. 2B) to display the effects of METH treatment in wild-type mice” and “In the METH-treated Tlr4-/- group, 125 upregulated and 182 downregulated genes were screened out compared to the METH-treated wild-type group (Fig. 6B) to demonstrate the effects of silencing the Tlr4 gene.” In addition, the definition of “FC” has been added to the Supplementary Tables. Mismatch of the supplementary tables with the table titles has been corrected.

2. Related, I’m having trouble with the authors’ use of the word, “enriched.” The authors need to be clear in their results whether transcripts are up- or downregulated between conditions and what particular data indicate these assertions.

Response 2: The “enriched” has been replaced by “identified” in the descriptions of Fig. 6.

3. Regarding the authors' response: “When Tlr4 was silenced, the inflammatory markers were suppressed-” where exactly is this shown in the RNA seq results? Figure 6 does not clearly indicate this assertion, nor does the description in the Results.

Response 3: The response of“When Tlr4 was silenced, the inflammatory markers were suppressed-” in R1 revision indicated that the mRNA and protein expression of MyD88, TRAF6, p65, IL-1β, and TNF-α were significantly suppressed in METH-treated Tlr4-/- mice compared to the METH-treated wild-type mice, as shown in Fig. 5. Consistently, the DEGs enriched in the Inflammatory mediator regulation of TRP channels and the Toll-like receptor signaling pathway (such as Tlr4, Tlr12, Camk2b, cyp4a14 etc.) were significantly downregulated. Therefore, the two pathways were significantly quenched by silencing the Tlr4 gene with the p value less than 0.001, as described in the Results text. The DEGs enriched in the Inflammatory mediator regulation of TRP channels and the Toll-like receptor signaling pathway in METH-treated Tlr4-/- mice compared to the METH-treated wild-type mice have been added as Supplementary Table 3.

4. I’m also having trouble with this statement: “In addition, inflammatory mediator regulation of TRP channels (p < 0.001), Toll-like receptor signaling pathway (p < 0.001), NF-κB signaling pathway (p = 0.082) and TNFsignaling pathway (p = 0.141) were enriched in METH-treated Tlr4-/- group through KEGG enrichment analysis compared to wild-type group (Fig. 6D), suggesting the regulation of inflammatory pathways by silencing Tlr4 in attenuating METH-induced hepatotoxicity in Tlr4-/- mice.” This statement still seems to imply that inflammatory pathways are upregulated (“enriched”) in Tlr4-/- mice. The word “regulation” in the phrase “suggesting the regulation of inflammatory pathways’” is vague. Something can be up- or downregulated. Please revise for clarity. Furthermore, some of these p-values do not reach a value of 0.05.

Response 4: The DEGs enriched in the Inflammatory mediator regulation of TRP channels and the Toll-like receptor signaling pathway (such as Tlr4, Tlr12, Camk2b, cyp4a14, etc.) were significantly downregulated, as shown in Supplementary Table 3. According to the suggestion, “suggesting the regulation of inflammatory pathways” has been replaced by “suggesting the quenched inflammatory pathways”. According to the instructions of the KEGG enrichment analysis, R script was used to conduct the analysis of the DEGs. A p value less than 0.05 was considered significant enrichment of the pathway. Although the NF-κB signaling pathway (p = 0.082) and TNF signaling pathway (p = 0.141) were listed in the results of KEGG enrichment analysis, the p-values did not reach a value of 0.05, suggesting no significant enrichment.

5. Abstract: “Moreover, inflammatory pathways were also significantly enriched in METH-treated Tlr4-/- mice compared to METH treated wild type mice suggesting the regulatory effect of Tlr4 silencing on inflammatory pathways.” This statement is confusing, and suggests to me that inflammatory pathways were upregulated in the Tlr4 knockouts, compared to the METH-treated wild type which seems contradictory. The authors have not adequately addressed this concern in either their response, or the revised manuscript. They need to elaborate on whether this is indeed the case. If it is, it would seem like a counterintuitive result that needs further explanation in the discussion section.

Response 5: According to the suggestion, the sentence has been rewritten, shown as “In addition, the dampening effects of silencing Tlr4 on inflammatory pathways were verified by the enrichment analysis of RNA-sequencing data in METH-treated Tlr4-/- mice compared to METH-treated wild-type mice.”

6. Page 14: This revision does not make sense, please revise: “consistent with our recent study, attributing to METH-caused the increased abundance of pathogenic gut microbiota”

Response 6: In the R1 revision, one of the other reviewers suggested us to insert our recent publication (Wang et al., 2022 doi:/10.1016/j.taap.2022.116011) and comment on the findings and highlights from that study in the concluding section of the manuscript. Therefore, the sentence was added.

7. Overall, there are still points that need to be clarified so that the reader can properly interpret the authors’ data and so that it is clear that their data support their conclusions. These are changes that were requested in the first review and have not been sufficiently addressed.

Response 7: We appreciate the time and effort the reviewer has put to give us such critical and constructive comments to improve the manuscript. We have performed the related suggestions the reviewers required and tried our best to address all concerns. We hope that we have answered and clarified all questions and comments to your full satisfaction and are looking forward to your reply.

Round 3

Reviewer 1 Report

No additional comments.